# A Case Study on Deformation Failure Characteristics of Overlying Strata and Critical Mining Upper Limit in Submarine Mining

Guang Li [1,2], Zhiwen Wang [1,2,3], Fengshan Ma [1,2,*], Jie Guo [1,2], Jia Liu [1,2,3] and Yewei Song [1,2,3]

1. Key Laboratory of Shale Gas and Geoengineering, Institute of Geology and Geophysics, Chinese Academy of Sciences, Beijing 100029, China
2. Institutions of Earth Science, Chinese Academy of Sciences, Beijing 100029, China
3. University of Chinese Academy of Sciences, Beijing 100049, China
* Correspondence: fsma@mail.iggcas.ac.cn; Tel.: +86-135-0107-2329

**Abstract:** Unlike land mining, the safety of seabed mining is seriously threatened by an overlying water body. In order to ensure the safety of subsea mining projects, it is of great importance to understand the failure characteristics and influencing factors of overlying strata deformation. Focusing on the Sanshandao Gold Mine, a typical submarine deposit in China, geomechanical model testing and numerical simulations were carried out. The results show that in the mining of a steeply dipping metal ore body, subsidence deformation mainly occurs on the hanging wall; the subsidence center is located on the surface of the hanging wall, and the uplift center is located on the upper surface of the ore body. The critical mining upper limit, which represents the minimum thickness of the reserved isolation pillar between the overlying seawater and the goaf, was determined to be 50 m in the Xinli mine; fault slip would occur if this critical value was exceeded. The dip angle and thickness of the ore body were negatively correlated with the vertical surface deformation. As the dip angle and thickness increased, the critical upper mining limit increased. When the fault was located in the footwall, the critical upper mining limit increased as the distance between the fault and the ore body increased, and the failure mode of the goaf was fault slip. When the fault was located in the hanging wall, the final failure mode of the goaf changed to a combined failure mode of overlying rock collapse as well as fault slip. These research results provide a theoretical basis for the selection of the reserved pillar height in the Xinli mining area, as well as a reference for safe mining practices under similar geological conditions.

**Keywords:** coastal mines; mining rock movement; failure characteristics of overlying strata; critical upper mining limit; similar model test

## 1. Introduction

With socio-economic development, the consumption of mineral resources has intensified, while deposits with better storage conditions have become nearly exhausted [1,2]. Consequently, the mining of deposits with poor geological storage conditions has been gaining more attention [3–5]. There is an abundance of unexploited mineral resources in the seafloor, and their potential applications are broad [6–9]. In contrast to developing land mines, submarine mine excavation is threatened by overlying water bodies. Moreover, metal mines often have a large dip angle, and the mining of steeply inclined ore bodies inevitably causes the working face at various depths to overlap in the vertical direction, making it easy for large-scale failure to occur under the mutual influence of the rock mass' deformation [10–12]. In addition, when the overlying surrounding rock undergoes a large amount of deformation, a water channel connecting with the seabed may form, resulting in seawater flooding into the mined-out area [13]. Seawater is not easy to drain, and its source is difficult to cut off; thus, it is of great importance for the safe and economic operation of

subsea mining projects to determine the critical thickness of the waterproof pillars left at the top of the mining area [14].

Undersea mining faces complex rock engineering geological problems, and the natural stress state of the bedrock becomes redistributed during human engineering activities. Stress concentration occurs in the goaf, which induces unloading rebound of the rock mass toward the mining area [15]. In order to solve these problems, scholars around the world have extensively studied the fracture mechanics, elastic–plastic mechanics, hydrogeology, hydrochemistry, and statistical mechanics, and a resulting series of achievements have been realized [16–20]. A combination of field monitoring, mechanical models, numerical simulations, and physical model tests is usually applied to research the rock movement and water intrusion that are induced by mining. However, the existing research results have not considered steep dip metal mines under the sea; thus, these issues need to be further studied. Moreover, in past studies, the selection of the waterproof pillar thickness has mainly been based on a generalized mechanical model and an empirical formula, and the impact assessment of specific geological environment conditions has been insufficient. Thus, few studies have been conducted on the factors that affect the critical upper mining limit.

Considering the issues mentioned above, the Sanshandao Gold Mine in the Xinli mining area, which is the only operating submarine mine in China, was taken as the research object. Based on the engineering geological conditions, physical model tests and numerical simulations were used to reproduce the filling mining process, in order to clarify the deformation failure characteristics of the surrounding rock of the mined-out area under dynamic mining conditions, to investigate the influences of the different geological conditions on the critical upper mining limit of the submarine mine, and to determine a critical upper mining limit for Xinli. These research results provide a theoretical basis for safe production in the Xinli mining area, and serve as a reference for mine development under similar geological conditions.

## 2. Study Area

The Sanshandao Gold Mine is the only coastal bedrock metal mine still being mined in China. It is located in the town of Sanshandao, Yantai City, Shandong Province (Figure 1a). It is surrounded by sea to the north, west, and south, and by land to the east [12]. The elevation of the mining area is 1.2–6 m, and the terrain is flat. The mining area is dominated by fault structures. Fault F1 is the ore controlling fracture, which has the greatest impact on mining. F1 has a water-repellent effect, and the thickness of the gouge is 0.05–0.5 m (Figure 1b). The fault zone strikes 35–40° NE and gradually steepens from NE to SW, with the average dip angle increasing from 46° to 70° (Figure 1c). The exposed strata in the area are mainly Quaternary sediments and granite of the Jiaodong Group. The bedrock of the deposit is mainly composed of metamorphic and magmatic rocks, and the ore body occurs in the alteration zone of fault F1. The alteration zone has been obviously zoned through multi-stage tectonic movement, and it basically spreads along the tendency and strike of fault F1. The gold deposits mainly occur in the cataclastic rocks within 40 m of fault F1, and they mainly consist of veined and reticulated mineralized rocks [21,22].

Horizontal tectonic stress plays a significant role in this area. The measured maximum principal stress value in the mining area can be as high as 35 MPa. The in-situ stress field in the study area is evenly distributed, without obvious mutation points. The measured values of the ground stress at different depths are presented in Table 1.

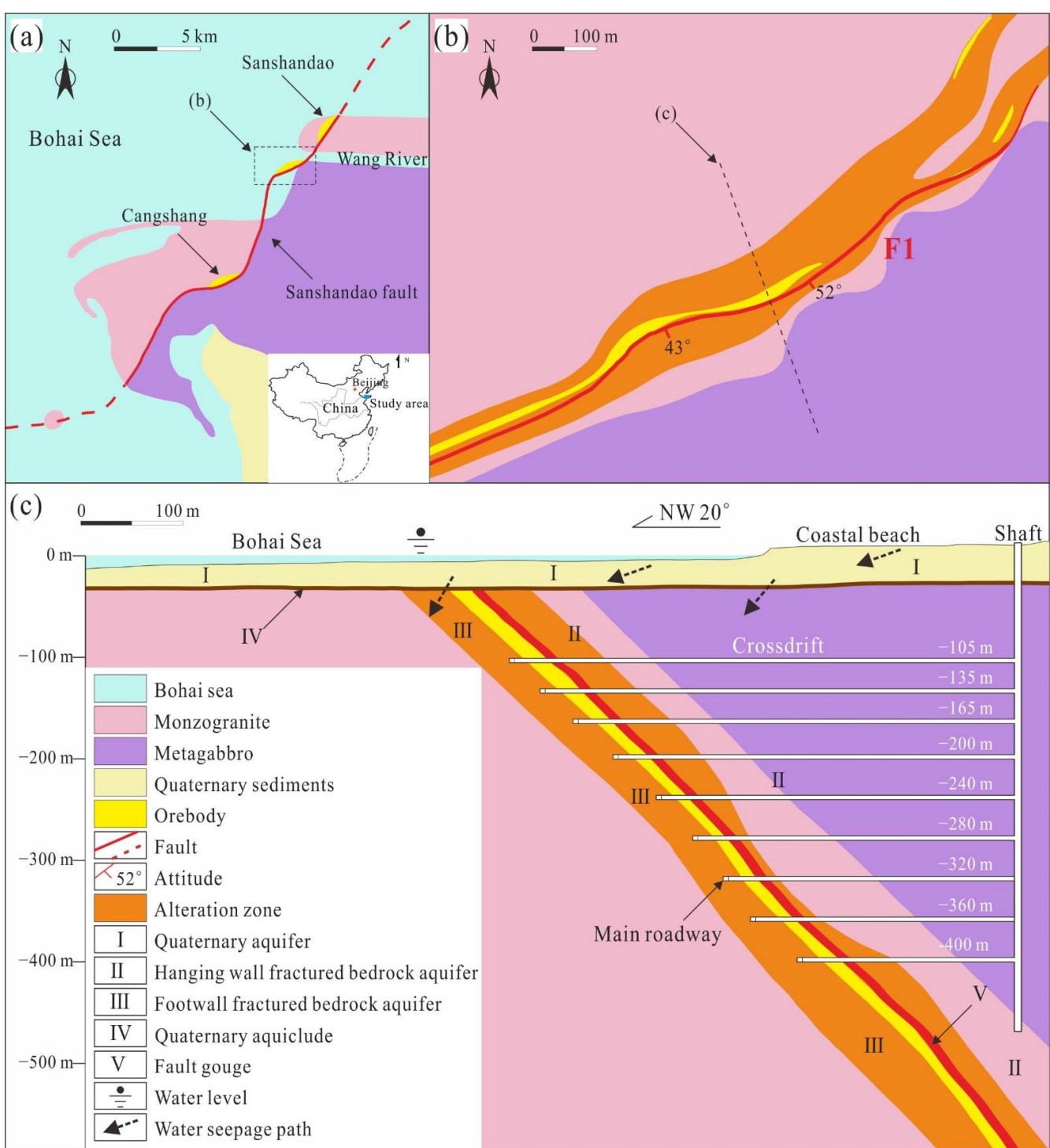

**Figure 1.** Engineering geological map of Sanshandao Mine. (**a**) Simplified geologic map showing the Sanshandao fault and other features; (**b**) close-up geologic map of the Xinli gold deposit area showing the fault and cross-section location; (**c**) cross-section showing the fault, geology, and hydrogeologic features, as well as the underground workings of the Xinli gold deposit.

**Table 1.** Ground stresses in the study area.

| Depth (m) | σ1 | | | σ2 | | | σ3 | | |
|---|---|---|---|---|---|---|---|---|---|
| | Value (MPa) | Direction (°) | Dip Angle (°) | Value (MPa) | Direction (°) | Dip Angle (°) | Value (MPa) | Direction (°) | Dip Angle (°) |
| 165 | 4.2 | 173.2 | 29.8 | 3.6 | 54.9 | 39.7 | 1.4 | 287.8 | 36.0 |
| 240 | 11.2 | 158.1 | 14.7 | 2.9 | 36.8 | 63.1 | −0.9 | 254.2 | 22.0 |
| 240 | 10.2 | 107.2 | 31.0 | 4.8 | 235.7 | 45.9 | −0.5 | 358.7 | 27.8 |
| 240 | 11.7 | 134.4 | 0.8 | 6.5 | 43.9 | 31.8 | 0.1 | 225.7 | 58.2 |
| 400 | 34.2 | 335.3 | 34.7 | 6.6 | 180.6 | 52.3 | 4.2 | 74.1 | 12.4 |
| 400 | 28.5 | 333.2 | 31.5 | 16.9 | 91.3 | 33.8 | 13.6 | 215.8 | 40.4 |

Based on least squares linear regression analysis, the variations in each principal stress value with burial depth can be obtained as follows:

$$\sigma_1 = 0.11 + 0.0539H \tag{1}$$

$$\sigma_2 = 0.13 + 0.0181H \tag{2}$$

$$\sigma_3 = 0.08 + 0.0315H \tag{3}$$

where $\sigma_1, \sigma_2,$ and $\sigma_3$ are the maximum principal stress, the intermediate principal stress, and the minimum principal stress (MPa), respectively, and $H$ is the depth (m).

The entire Xinli mining area is located below sea level, with overlying seawater, and there is no natural drainage occurring during the mining process. The Quaternary water-bearing strata are connected with the mined-out area, and the ore-controlling fault has rich supply sources. There is only one subclay water-resistant layer between the mined-out area and fault F1. Under mining activities, the top water-resistant layer may be broken as a result of rock movement, and fault F1 could slip into a water-conducting channel, which poses a threat to the mining. The upward horizontal slicing and filling mining methods are applied, the full tailings are used for backfilling, and a combination of rock drilling and blasting are used for mining. Since 2005, the Xinli deposit has been mined down to −165 m, and a 130-meter-thick waterproof pillar has been retained at the top. At present, the bottom ore body is nearly exhausted, and the mining experience has proven that the thickness of the top waterproof pillar is conservative, which wastes part of the resources. Thus, it is necessary to determine a reasonable critical upper mining limit based on the geological conditions of the mining area.

## 3. Physical Model Tests

Physical model tests are an important means for studying complex engineering geological problems because the physical model can simulate the deformation from elastic to plastic, and can replicate the entire destruction process in a laboratory experiment [23].

### 3.1. Simulation Prototype

In the design of the physical model, it is very complicated to fully realize a fine characterization of the mine's structure, its geo-stresses, and hydrology. Therefore, the engineering prototype was simplified appropriately according to the main issues being studied, and a typical section was selected (Figure 1c). The section was generalized according to a lithological association with the rock mass' structure. Based on comprehensive considerations of the laboratory conditions as well as the testing equipment, the upper thickness of the ore body was set at 40 m; the lower thickness was set as 73 m; the dip angle was set at 45°; and the upper boundary was approximately tangential to the ore-controlling fault F1 (Figure 2). The thickness of the Quaternary sediments on top of the rock mass was set to approximately 35 m.

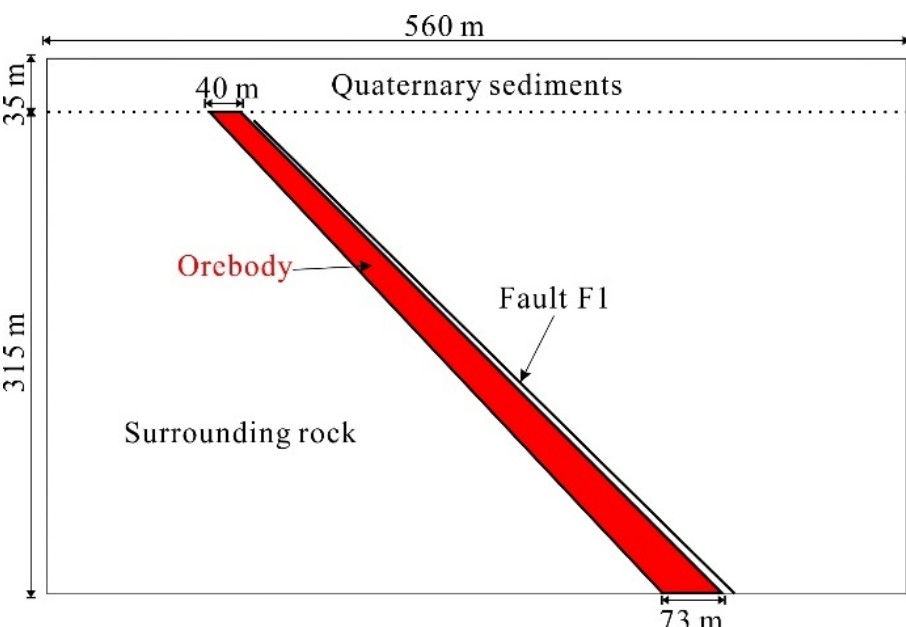

**Figure 2.** Simplified geologic profile of the study area.

*3.2. Similarity Relationship*

Physical experiments require sufficient similarity to be established between the model and the prototype. However, because of the high demands on the materials, equipment, and technology, it is difficult to achieve complete similarity. Therefore, several important indicators were selected according to the goals of the study. The dimensionless ratio of the same quantities in the prototype and model was used as the similarity constant, C. Gravity is an important force in various physical processes; thus, it was necessary to achieve sufficient density similarity between the similar materials. In the study of mining stability, the compressive strength and elastic modulus are indispensable indexes that ensure that the deformation failure characteristics of the model and prototype are sufficiently similar during the excavation process. Firstly, based on the laboratory conditions, the geometry similarity constant $C_l$ was selected. Then, other basic physical quantities were calculated in accordance with the law of Buckingham $\pi$ theorem, as shown in Table 2.

**Table 2.** Physical and mechanical parameters of the rock and rock mass.

| Physical Quantity | Similarity Relationship | Similarity Constant |
|---|---|---|
| Geometry (key constant) | $C_l$ | 350 |
| Density (key constant) | $C_\rho$ | 1.3 |
| Displacement (key constant) | $C_D$ | 350 |
| Poisson's ratio | $C_\mu = 1$ | 1 |
| Elastic modulus | $C_E = C_\rho C_l$ | 455 |
| Strain | $C_\varepsilon = C_\rho C_l / C_E$ | 1 |
| Stress | $C_\sigma = C_l C_\gamma$ | 455 |
| Internal friction angle | $C_\varphi = 1$ | 1 |
| Cohesion | $C_c = C_\rho C_l$ | 455 |

*3.3. Similar Materials*

A crucial part of a large-scale physical model test is the fast and accurate determination of the ratio of the similar materials. In this experiment, river sand was used as the aggregate; high-strength gypsum and ordinary Portland cement were used as the cementing agents; and an accelerating agent was used as an additive to make similar materials. A standard specimen with a diameter of 50 mm and a height of 100 mm was constructed. The density of the specimen was measured, and a uniaxial compression test was carried out. Similarly,

in the material proportioning test, the parameters of the similar materials can be changed by adjusting the proportional relationships among the river sand, cement, and gypsum. The experimental results indicated that the river sand:cement:gypsum ratios of the matching ore rock, surrounding rock, and Quaternary sediments are 50:2:3, 20:1:1, and 65:2:3, respectively. The physical and mechanical parameters of the rock mass and similar material are presented in Table 3.

**Table 3.** Physical and mechanical parameters of the rock mass and similar material.

| Type | Lithology | Density (g·cm$^3$) | Compressive Strength (MPa) | Elastic Modulus (MPa) | Poisson's Ratio |
|---|---|---|---|---|---|
| Measured value | Ore rock | 2710 | 60 | 2100 | 0.32 |
| | Surrounding rock | 2650 | 58 | 7300 | 0.26 |
| | Quaternary sediments | 1750 | 8 | 150 | 0.22 |
| Design value | Ore rock | 2084 | 0.13 | 5 | 0.32 |
| | Surrounding rock | 2038 | 0.13 | 16 | 0.26 |
| | Quaternary sediments | 1346 | 0.02 | 0.33 | 0.22 |
| Similar material | Ore rock | 1875 | 0.132 | 6.6 | 0.29 |
| | Surrounding rock | 1889 | 0.167 | 14.3 | 0.25 |
| | Quaternary sediments | 1676 | 0.01 | 0.33 | 0.23 |

### 3.4. Loading and Monitoring

The model was loaded using a self-developed hydraulic servo comprehensive experimental platform consisting of three parts: a model box, a loading system, and a control system. The loading system was controlled by a computer, with a maximum loading force of 300 kN in the horizontal and vertical directions [5]. In this test, a geological body at a depth of 350 m below the surface was simulated. Due to limitations of the experimental conditions, gradient loading could not be realized in the horizontal direction. Therefore, the horizontal ground stress at a burial depth of 175 m was taken as the loaded ground stress in the test. According to the measured in-situ stress in the study area, the horizontal load that needed to be applied in the laboratory was 4.2 kN based on the similarity conversion.

The experiment was recorded using two digital cameras. One camera captured photos at regular intervals, while the other recorded the entire process. An optical speckle measurement system was used to observe model deformation during the excavation, and GOM Correlate software was used to interpret the images. The digital image correlation method compares several digital images before and after specimen deformation, and obtains the deformation information for the monitored area by calculating the displacement of a given point, which has the advantages of full-field measurement, a strong anti-interference ability, and high measurement accuracy.

### 3.5. Experimental Process

Based on the actual mining process in the study area, filling mining was initially carried out in three sections of the ore-body below −165 m; that is, at depth ranges of −200 m to −165 m, −240 m to −205 m, and −280 m to −245 m. The displacement of the surrounding rock was monitored during the mining. Then, the reserved water-proof pillars above −165 m were gradually mined and filled. The critical height when the water channel formed was obtained, and subsequently, the critical upper limit of mining was determined.

During ore body mining, the model was initially subjected to a horizontal load in order to simulate the horizontal in-situ stress; then, mining was carried out in each section after the load was stabilized. After the mining in each section was completed, the mined-out area was backfilled with mining residue mixed with cement and gypsum as filling materials. Taking the testing cost and equipment conditions into consideration, the mining time interval for each section was set at 1 h. The specific mining and filling conditions of each section are shown in Figure 3.

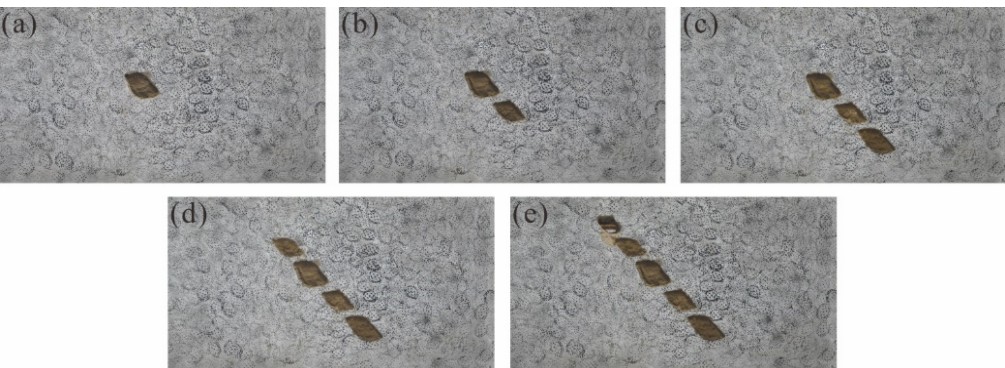

**Figure 3.** Physical model of the test process. (**a**) Mining and filling between −200 m and −165 m; (**b**) mining and filling between −240 m and −205 m; (**c**) mining and filling between −280 m and −245 m; (**d**) mining and filling between −160 m and −125 m; (**e**) mining and filling between −120 m and −85 m.

*3.6. Analysis of Physical Simulation Results*

3.6.1. Deformation Failure Characteristics of Overlying Strata

The speckle sprayed on the surface of the model was affected by the exfoliation of the model's skin. Therefore, seven survey lines with clear speckles were selected on the model for data analysis, and the series of curves obtained are shown in Figure 4.

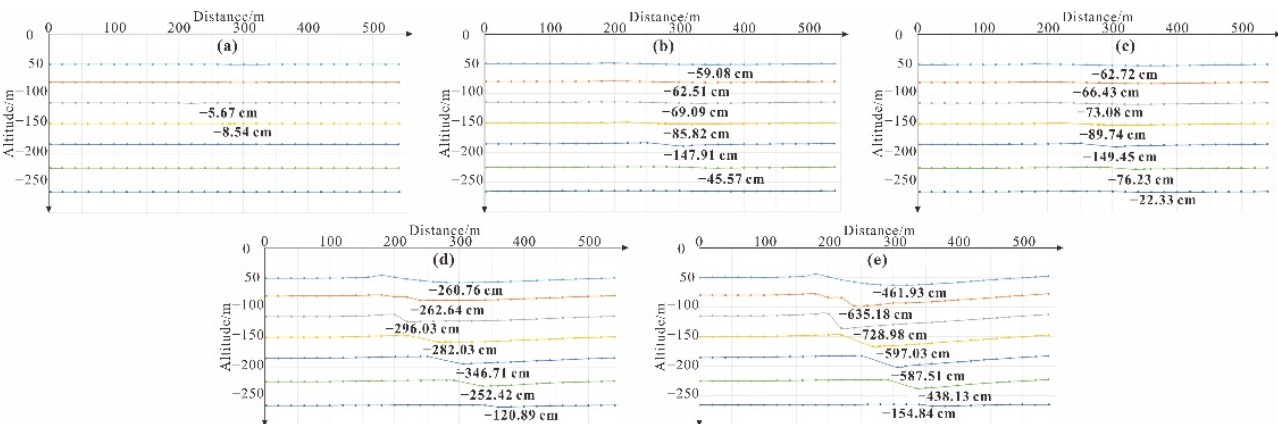

**Figure 4.** Displacement of surrounding rock during the entire mining process. (**a**) Mining and filling between −200 m and −165 m; (**b**) mining and filling between −240 m and −205 m; (**c**) mining and filling between −280 m and −245 m; (**d**) mining and filling between −160 m and −125 m; (**e**) mining and filling between −120 m and −85 m.

As shown in Figure 4a, during the mining from −200 m to −165 m, the goaf roof and hanging wall ore bodies became deformed. The maximum deformation occurred on the ore body between the goaf and the fault, and the deformation was about 8.54 cm. The deformations of the goaf floor and footwall were not obvious. Figure 4b shows that during the mining from −240 m to −205 m, the downward displacement of the hanging wall area increased significantly. The maximum deformation point was located between the upper right side of the section and the fault, and the maximum deformation detected was 147.91 cm. The surrounding rock of the lower left side of the lozenge goaf experienced an unloading rebound and obvious vertical displacement. As the mining deepened, the historical goaf was located on top of the existing goaf. The original goaf floor was transformed into the existing goaf roof, and the deformation of the top gradually accumulated. The stress state of the rock layer at the top changed to a cantilever beam type, and the rock mass in the hanging wall area underwent continuous and uniform bending subsidence. As shown in Figure 4c, when the working face advanced to the −280 m to

−245 m section, the deformation of the goaf hanging wall became funnel-shaped. The maximum deformation was 149.45 cm, and it occurred in the upper right part of the mining section, which may have been affected by the repeated mining activity.

After the downward mining was completed, the mining activity was changed to top ore-body stoping. As shown in Figure 4d, during the mining from −160 m to −125 m, it was found that except for the deformation of the roof of the existing goaf, the deformation of the historical goaf increased synchronously, and the maximum deformation point was still located in the second section of the downward mining. This may be due to the weakening of the rock's bearing capacity near the ore body as cumulative mining increased. Under the action of the right boundary load, the hanging wall rock mass moved toward the mined-out filling area, which had a low rock mass strength, causing deformation of the deep historical goaf to continue to increase as the shallow ore body was exploited. As can be seen from Figure 4e, during the mining from −120 m to −85 m, the surrounding rock of the footwall was uplifted, the settlement became severe in the hanging wall area, and the surrounding rock underwent differential deformation along the fault.

3.6.2. Critical Upper Mining Limit

For the mining process, the surrounding rock mass was divided into a collapse zone, yield zone, elastic deformation zone, and undisturbed zone, according to the distance from the stope. The rock mass in the collapse zone separated from the matrix and lost its bearing capacity. The integrity of the rock mass in the yield zone was damaged, and the rock mass had not yet separated from the matrix; however, the mass easily transformed into a collapse zone under mining activity. The elastic deformation zone was the farthest from the excavation space, and only elastic deformation of the surrounding rock mass occurred. With the expansion of the mining face, the existing mined-out area was affected by the historical area, and its plastic deformation area was larger than the single mined-out area. In addition, the movement range of the rock mass was also larger, the amount of collapse increased, and the deformation of the rock mass was aggravated. This process reduced the bearing capacity of the surrounding rock and caused deformation of the surface. First, a tensile crack initiated in the top of the goaf (Figure 5a); then, the crack expanded and cut the rock mass at the top (Figure 5b), causing a wedge to collapse from the overlying rock mass (Figure 5c). The quality of the surrounding rock decreased, and tie rock on both sides of the fault experienced staggered slip (Figure 5d).

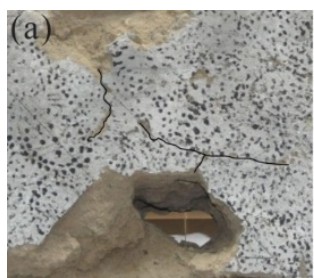 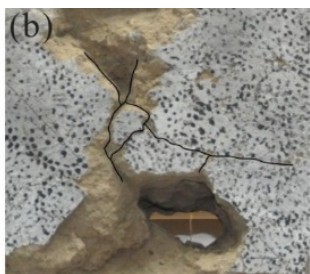 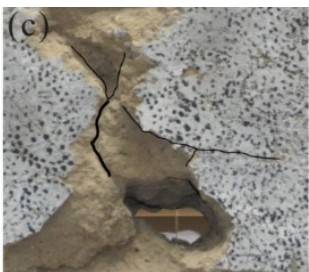 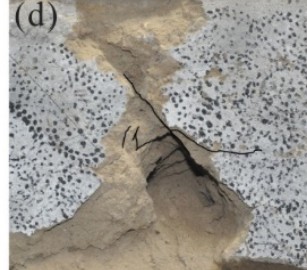

**Figure 5.** The entire process of the deformation and failure of the overlying strata. (**a**) Tensile cracks on the top of the goaf; (**b**) Cracks expanding; (**c**) Deterioration of the cracks; (**d**) Roof caving.

The stress field in the rock mass was in equilibrium before being disturbed by mining. A caving area formed inside the rock mass during excavation, which provided space for the deformation of the surrounding rock. In addition, under the action of the stress field, the surrounding rock experienced unloading rebound toward the goaf, and the stresses on part of the rock mass became concentrated due to the redistribution of the stress. Failure occurred when the rock mass' bearing limit was exceeded, and a new equilibrium state was achieved through stress transfer. In the process of mining from deep to shallow in the study area, the volume of the goaf increased, the strength of the surrounding rock decreased, and the fault slip resistance decreased. In the mining of the ore bodies above

−120 m, the upper wall's rock mass in the shallow part was less affected by gravity, and an obvious climbing effect was produced along the fault under the action of the horizontal stress. The tensile stress concentration zone was formed on the upper wall of the shallow goaf, and a horizontal crack was observed. The hanging wall of the deep goaf was mainly affected by gravity, and it mainly moved toward the goaf and underwent a small amount of displacement along the fault.

The physical simulation results revealed that as the mining and filling continued upwards from −85 m, the surrounding rock could not reach a new equilibrium state under the redistribution of the stress, and the surrounding rock consequently became unstable, finally leading to loss of the roof's bearing capacity and the formation of a penetrating crack between the goaf and the Quaternary deposit. At this time, the critical mining height was −85 m, and the thickness of the reserved pillar on the top was 50 m. The failure mode between the goaf and the uppermost aquifer was fault slip, and significant dislocation occurred along the fault, forming a channel for seawater intrusion.

## 4. Numerical Simulations

In order to further study the factors influencing the stability of seabed mining, as well as to verify the results of the physical model test, numerical simulations based on the discrete element method (DEM) were performed. In contrast to the finite element method (FEM), which generalizes the rock mass into a continuum of media, the interactions between the elements in the DEM can reflect the discontinuity of the rock mass, which conveniently describes their nonlinear mechanical behavior [24]. In addition, large deformations and fractures were allowed, which can better simulate the fracturing of the rock mass and slip along the structural plane. In this experiment, the particle discrete element software PFC (Particle Flow Code), which is mature technology in China and abroad, was used for the modeling. Its usage made it convenient to observe the initiation, expansion, and coalescing of the cracks when deformation and failure occurred near the goaf, which was crucial to obtaining the critical mining height [25,26].

### 4.1. Numerical Simulation Model

Based on the engineering geological conditions of the study area, the model was 560 m long and 350 m high. From top to bottom, the model was composed of Quaternary sediments, the bottom water-resistant layer, and the bedrock layer (hanging wall surrounding rock, ore body, and footwall surrounding rock, respectively). The particle size was 0.9–1.0 m, and the ball number was 66371. Fixed boundary conditions were applied to the bottom of the model, the gradient of the ground stress was applied on both sides, and the top was set as a free surface. The parallel bond model, which can transfer force and torque, was selected as the contact constitutive model, and the displacement and crack propagation were monitored using a program based on FISH language [27]. The numerical simulation model is shown in Figure 6.

Since the microscopic mechanical parameters are used in the PFC numerical calculation to reflect the macroscopic mechanical properties of the rock and soil materials, it was not possible to use the macroscopic mechanical parameters of the rock mass in the mining area. A calibration model was thus established according to the macroscopic mechanical parameters, in order to calibrate the microscopic mechanical parameters of the materials. The rock mass parameters measured in the Sanshandao mining area were used to calibrate the micro-mechanical parameters through uniaxial compression tests [27]. The micro-mechanical parameters obtained are presented in Table 4 [28].

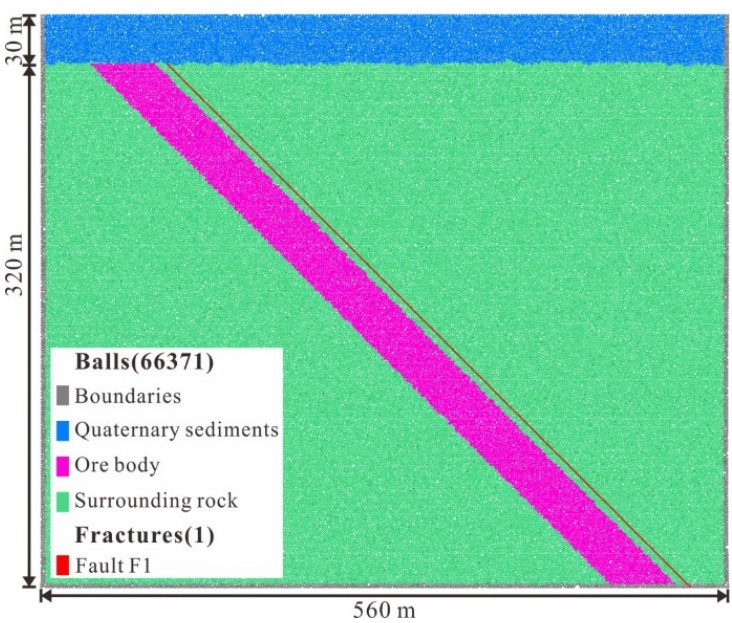

**Figure 6.** The numerical simulation model.

**Table 4.** Microscopic parameters in PFC.

| Lithology | Type | Parameter | Magnitude | Parameter | Magnitude |
|---|---|---|---|---|---|
| Ore rock | Particles | Density (kg/m$^3$) | 2700 | Young's modulus (GPa) | 1.72 |
| | | Minimum particle radius (mm) | 60 | Ratio of normal to shear stiffness | 1.84 |
| | | Ratio of maximum to minimum particle radius | 1.2 | Friction coefficient | 0.577 |
| | Parallel Bond | Young's modulus (GPa) | 1.55 | Cohesion (MPa) | 20 |
| | | Ratio of normal to shear stiffness | 1.72 | Internal friction angle (°) | 25 |
| | | Tensile strength (MPa) | 20 | Bond radius multiplier | 1.5 |
| Surrounding rock | Particles | Density (kg/m$^3$) | 2600 | Young's modulus (GPa) | 2.08 |
| | | Minimum particle radius (mm) | 60 | Ratio of normal to shear stiffness | 2.5 |
| | | Ratio of maximum to minimum particle radius | 1.2 | Friction coefficient | 0.5 |
| | Parallel Bond | Young's modulus of particle (GPa) | 2.08 | Cohesion (MPa) | 15 |
| | | Ratio of normal to shear stiffness | 2.5 | Internal friction angle (°) | 25 |
| | | Tensile strength (MPa) | 10 | Bond radius multiplier | 1.5 |

**Table 4.** *Cont.*

| Lithology | Type | Parameter | Magnitude | Parameter | Magnitude |
|---|---|---|---|---|---|
| Quaternary sediments | Particles | Density (kg/m$^3$) | 1750 | Young's modulus (GPa) | 1.55 |
| | | Minimum particle radius (mm) | 60 | Ratio of normal to shear stiffness | 2.5 |
| | | Ratio of maximum to minimum particle radius | 1.2 | Friction coefficient | 0.5 |
| | Parallel Bond | Young's modulus of particle (GPa) | 1.55 | Cohesion (MPa) | 1 |
| | | Ratio of normal to shear stiffness | 2.5 | Internal friction angle (°) | 20 |
| | | Tensile strength (MPa) | 0.2 | Bond radius multiplier | 1.2 |

The horizontal gradient stresses were realized by setting multiple sections of the wall on the lateral boundary of the model, and providing different servo stresses to the walls. Since the height of the model was 350 m and the calculation efficiency was limited, 35 sections of 10-meter-high walls were set on the left and right boundaries. According to the depth of the wall, the measured mean ground stress in the study area was applied. Based on the PFC, three influencing factors, namely the ore-body thickness, ore body dip angle, and fault location, were studied, and 11 models were constructed according to the experimental scheme presented in Table 5.

**Table 5.** Experimental scheme.

| | 1 | 2 | 3 | 4 | 5 | 6 | 7 | 8 | 9 | 10 | 11 |
|---|---|---|---|---|---|---|---|---|---|---|---|
| Dip angle | 45° | 55° | 65° | 75° | 45° | 45° | 45° | 45° | 45° | 45° | 45° |
| Thickness | 40 m | 40 m | 40 m | 40 m | 20 m | 40 m | 60 m | 40 m | 40 m | 40 m | 40 m |
| Fault location | Hanging wall | Hanging wall | Hanging wall | Hanging wall | Hanging wall | Hanging wall | Hanging wall | Foot wall | Foot wall | Hanging wall | Hanging wall |
| Fault-ore distance | 5 m | 5 m | 5 m | 5 m | 5 m | 5 m | 5 m | 30 m | 5 m | 5 m | 30 m |

### 4.2. Analysis of Models with Different Dip Angles

The dip angle of the ore body in the Xinli mining area is between 40° and 80°. In order to study the influence of the dip angle of the ore body on the deformation and failure of the surrounding rock during mining, the angle was set at 45°, 55°, 65°, and 75° in the filling mining tests. According to the mining practice and physical model test results, mining was carried out below −170 m, and then, mining was carried out above −165 m. After reaching −85 m, the mining scale was reduced, and mining was continued upward at a rate of 5 m each time until failure occurred at the top. After failure occurred, we continued to calculate 4000 steps in order to obtain the deformation failure characteristics of the surrounding rock in more detail. The simulation stages are shown in Table 6.

**Table 6.** Simulation calculation stages.

| Stage | 1 | 2 | 3 | 4 | 5 | 6 |
|---|---|---|---|---|---|---|
| Mining zone | −200 m to −170 m | −240 m to −205 m | −280 m to −245 m | −165 m to −135 m | −130 m to −100 m | −95 m to −90 m |
| Stage | 7 | 8 | 9 | 10 | 11 | 12 |
| Mining zone | −90 m to −85 m | −85 m to −80 m | −80 m to −75 m | −75 m to −70 m | −70 m to −65 m | −65 m to −60 m |

### 4.2.1. Surface Displacement

Measuring circles were set in the model to monitor the vertical displacement on the surface during the excavation and filling. The results are shown in Figure 7. Figure 7 shows that during mining, the subsidence center formed on the upper wall of the surface, and the uplift center formed on the surface near the top of the ore body. As the dip angle increased, the maximum vertical subsidence of the surface decreased, and the subsidence curve became smooth. When the dip angle of the ore body was small, only one single settlement center formed on the hanging wall. As the dip angle increased, another small settlement center formed on the surface due to the collapse of the top of the ore body. In addition, the horizontal position of the maximum surface subsidence point moved away from the ore body as the dip angle increased.

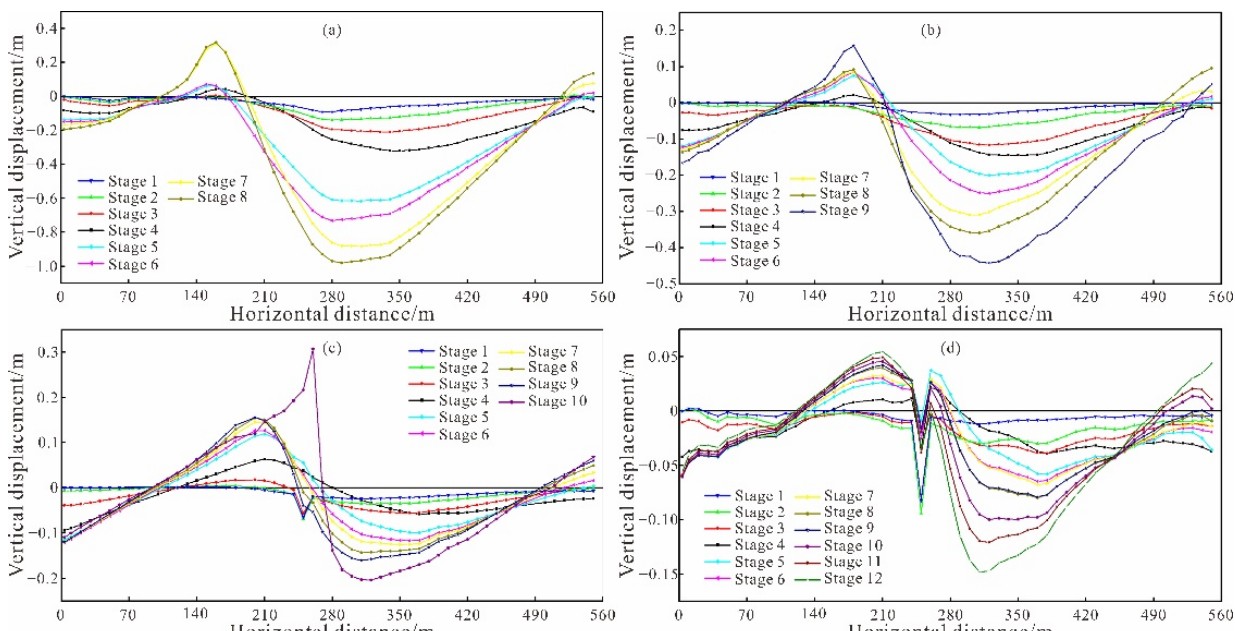

**Figure 7.** Physical model test process. (**a**) Dip angle of 45°; (**b**) dip angle of 55°; (**c**) dip angle of 65°; (**d**) dip angle of 75°.

### 4.2.2. Failure Mode

It can be seen from Figure 8 that the dip angle of the ore body had a significant influence on the failure mode and on the critical upper mining limit. As the dip angle increased, the overall settlement of the ore body hanging wall decreased, and the thickness of the isolated pillar on the top of the goaf decreased. Furthermore, the failure mode changed from fault slip to roof collapse. The reason for this may be that when the ore body's dip angle was small, the slip resistance of the fault decreased rapidly as the mining section increased; the fault was activated, the hanging wall rock mass climbed along the fault, and failure occurred even though the top isolation pillar was thick. When the dip angle of the ore body was large, the anti-sliding force provided by the rock mass on both sides of the fault was sufficient to inhibit fault slip; thus, the critical upper mining limit of the steeply inclined ore body was greatly increased.

### 4.3. Analysis of Models with Different Ore-Body Thicknesses

The ore-body thickness in the Xinli mining area ranges from 20 m to 60 m; thus, the ore-body thickness was set at 20 m, 40 m, and 60 m in the numerical models, and the same excavation-filling sequence was used in the simulation calculations.

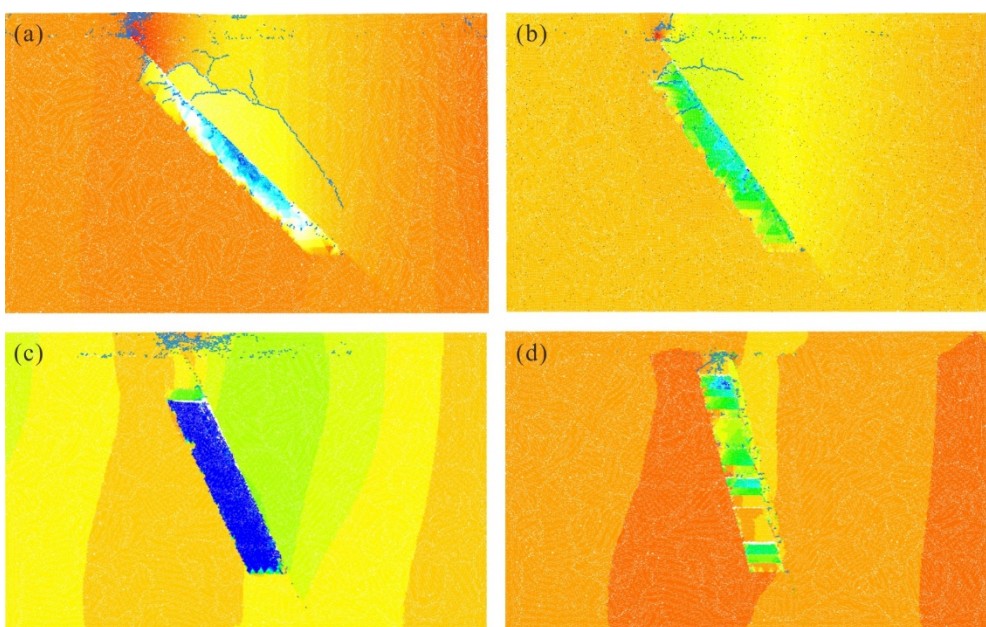

**Figure 8.** Failure modes for the mining of an ore body with different dip angles. (**a**) Dip angle of 45°; (**b**) dip angle of 55°; (**c**) dip angle of 65°; (**d**) dip angle of 75°.

### 4.3.1. Surface Displacement

As shown in Figure 9, as the mining progressed, an uplift center formed on the surface near the top of the ore body, and a settlement center formed on the surface of the hanging wall. As the thickness of the ore body increased, the maximum surface settlement decreased, and the surface settlement curve became smooth. The horizontal position of the maximum surface settlement point moved toward the hanging wall, and the maximum displacement of the surface uplift moved closer to the direction of the central ore body in the model as the thickness of the ore body increased.

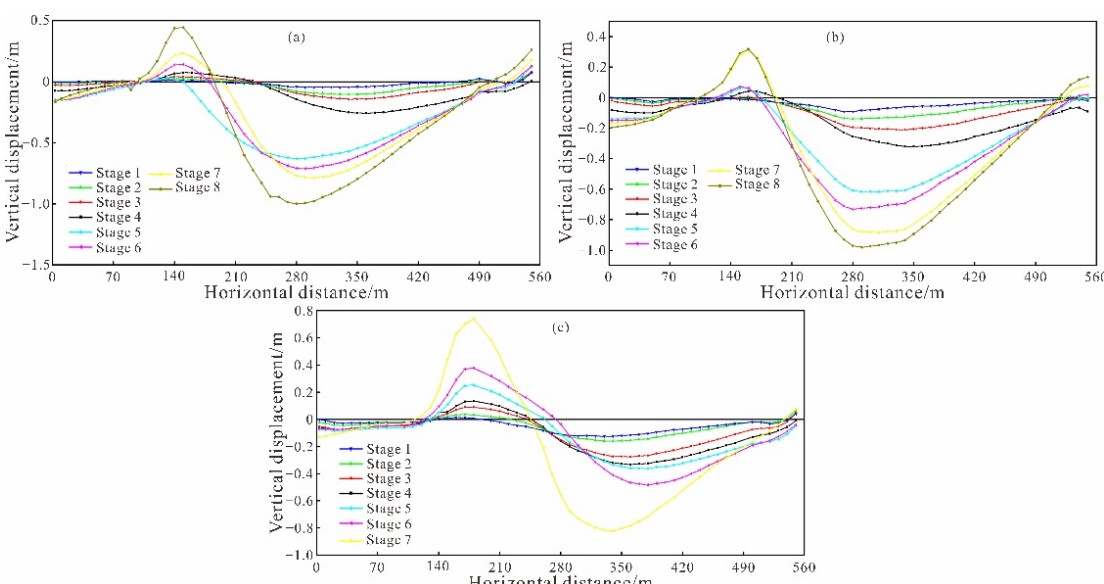

**Figure 9.** Surface vertical displacement of an ore body with different thicknesses. (**a**) Thickness of 20 m; (**b**) thickness of 40 m; (**c**) thickness of 60 m.

### 4.3.2. Failure Mode

It can be seen from Figure 10 that the ore body's thickness had an obvious influence on the critical upper mining limit, but it did not have an obvious influence on the failure mode

of the ore body. As the thickness of the ore body increased, the overall settlement on the hanging wall increased, and the height of the isolated pillar on the top of the goaf decreased. When the top isolation pillar was excavated, the degree of damage to the surrounding rock was aggravated; finally, fault slip was induced, and a water channel was formed.

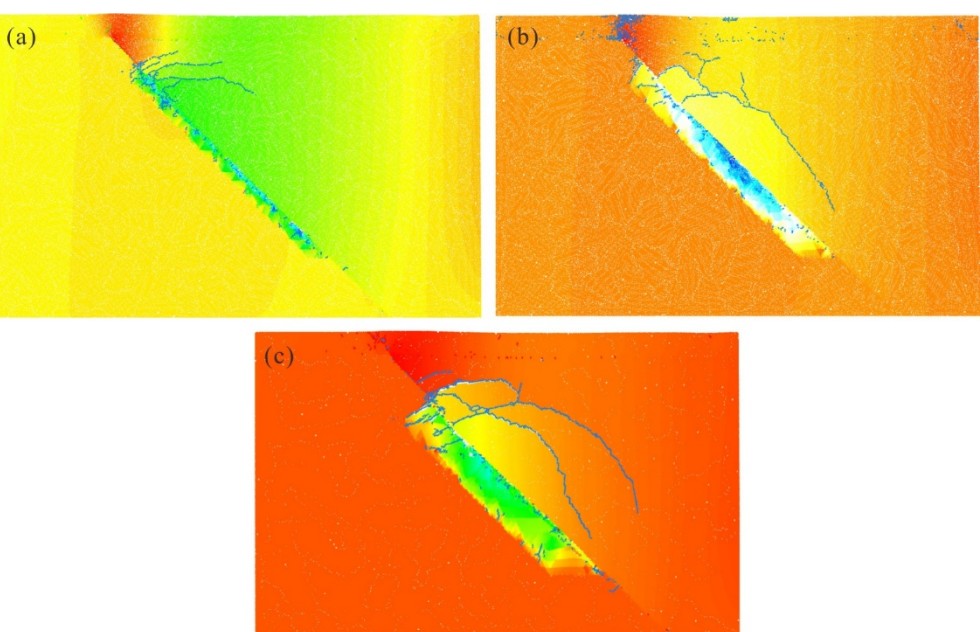

**Figure 10.** Failure modes for mining of an ore body with different thicknesses. (**a**) Thickness of 20 m; (**b**) thickness of 40 m; (**c**) thickness of 60 m.

*4.4. Analysis of Models with Different Fault Locations*

The Xinli mining area is controlled by a single fault (F1), and the distance between the ore body and the fault differs between sections. Based on the actual geological conditions, four fault locations, including 30 m in the hanging wall, 5 m in the hanging wall, 5 m in the foot wall, and 30 m in the foot wall, were simulated.

4.4.1. Surface Displacement

The vertical surface displacements that were obtained based on the measurement circle are shown in Figure 11. When the fault was located below the ore body, the maximum surface settlement increased slightly as the distance between the ore body and the fault increased; the settlement center moved horizontally toward the hanging wall. The maximum surface uplift increased slightly, and the uplift center moved away from the foot wall. When the fault was located on the upper side of the ore body, the maximum surface subsidence center increased significantly as the distance between the ore body and the fault increased. The vertical displacement of the surface uplift center increased, and the vertical displacement curve tended to become smooth. Compared to when the fault was located in the hanging wall, when the fault was located in the foot wall, the displacement of the surface subsidence center increased, and the displacement of the surface uplift center decreased.

4.4.2. Failure Mode

It can be seen from Figure 12 that the fault location had an obvious influence on the failure mode and on the critical upper mining limit. When the fault was located in the foot wall, as the distance between the fault and the ore body increased, the critical upper mining limit increased, and the thickness of the pillar left on the top decreased. Multiple horizontal cracks formed due to the instability of the surrounding rock between the goaf and the fault, and fault activity was consequently triggered. Then, a water channel formed between the

goaf and the Quaternary aquifer along the fault. When the fault was located in the hanging wall, the final failure mode of the goaf changed to a combined failure mode of overlying rock collapse and fault slip. As the distance between the fault and the ore body increased, the impact of the collapse of the rock mass in the top isolation layer became more severe. Compared with when the fault was located in the hanging wall, when the fault was located in the foot wall, the critical upper mining limit increased, and the thickness of the water barrier pillar decreased.

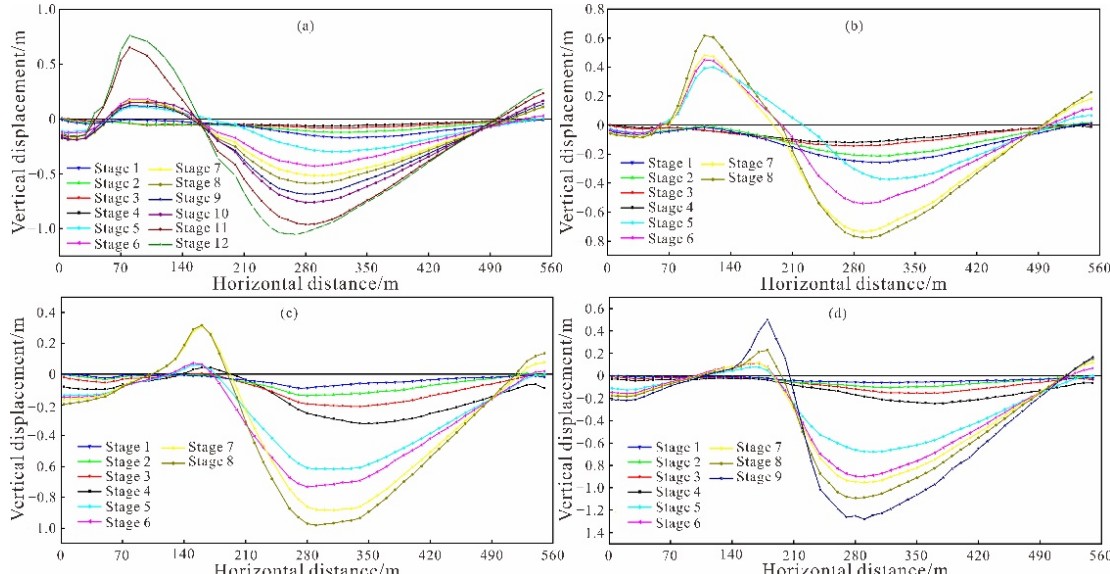

**Figure 11.** Surface vertical displacement of an ore body with different fault locations. (**a**) Fault location at 30 m in the foot wall; (**b**) fault location at 5 m in the foot wall; (**c**) fault location at 5 m in the hanging wall; (**d**) fault location at 30 m in the hanging wall.

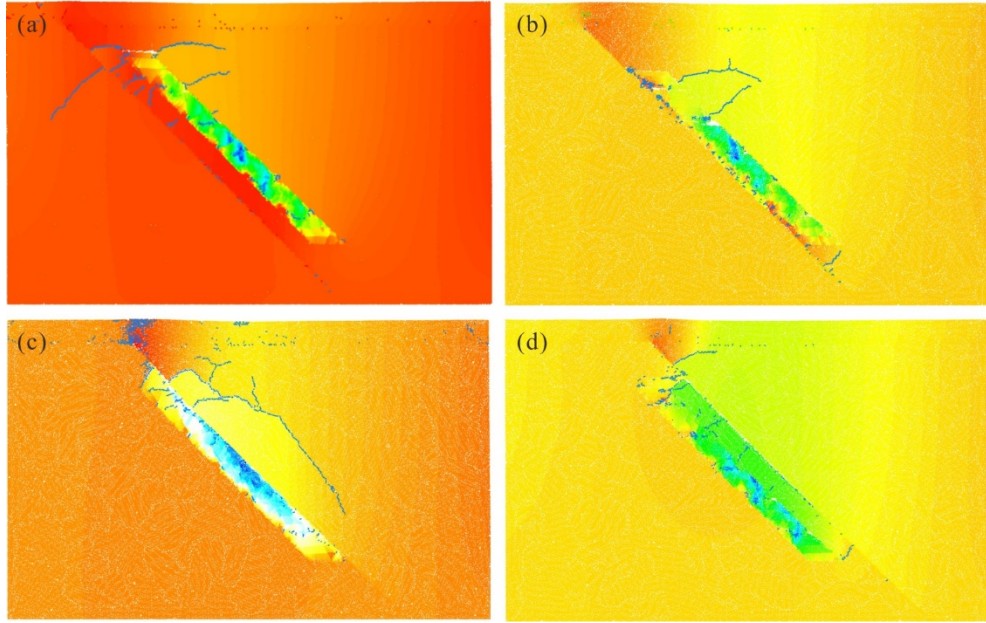

**Figure 12.** Failure modes for mining of an ore body under different fault locations. (**a**) Fault location at 30 m in the foot wall; (**b**) fault location at 5 m in the foot wall; (**c**) fault location at 5 m in the hanging wall; (**d**) fault location at 30 m in the hanging wall.

## 5. Discussion

Most metal ore bodies have a large dip angle, which leads to the superposition of the vertical displacement on the goaf, forming a large deformation zone on the surface. When the differential settlement reaches a certain degree, a tensile stress concentration zone is formed on the top of the excavation area, which leads to collapse. The impact of this disaster on seabed mining is more serious. Excessive mining may form a water-conducting fracture zone that cuts through the seabed's waterproof layer and induces water inrush. The dip angle and thickness of the ore body have significant influences on the vertical displacement of the surface, while the fault position has little influence on it. As the dip angle increases, the maximum vertical settlement of the surface decreases, and the maximum settlement point moves horizontally away from the ore body. As the thickness of the ore body increases, the maximum vertical settlement of the surface decreases, and the surface settlement curve becomes moderate. The horizontal position of the maximum surface subsidence point moves upward toward the hanging wall as the thickness of the ore body increases.

In order to reduce the hidden danger of underwater water inrush, it is necessary to reserve a certain thickness of ore body on the top of the mining area as a safety isolation layer. The selection of the isolation layer pillar's thickness determines the critical upper mining limit, which is directly related to the economic benefit and safe production of the mine. The dip angle, the thickness of the ore body, and the fault location all have influences on the critical upper mining limit. As the dip angle of the ore body increases, the overall settlement of the hanging wall of the ore body decreases, the thickness of the isolated pillar decreases, and the critical upper mining limit increases. As the thickness of the ore body increases, the overall settlement on the hanging wall of the ore body increases, and the thickness of the isolated pillar decreases. When the fault is located in the foot wall, as the distance between the fault and the ore body increases, the critical upper mining limit increases, and the thickness of the pillar left on the top decreases. Compared with when the fault is located in the hanging wall, when the fault is located in the foot wall, the critical upper mining limit increases.

This study summarizes the penetrative failure modes of a submarine metal mine, and provides a basis for mine risk assessment. It also provides important guidance for mine disaster prevention and emergency treatment. The dip angle and fault location of the ore body can change the penetrative failure mode between the overburden and ore body, but the thickness of the ore body has little influence on it. As the dip angle of the ore body increases, the failure mode changes from fault slip to roof collapse. When the fault is located in the footwall, the failure mode of the goaf is fault slip. When the fault is located in the hanging wall, the final failure mode of the goaf changes to a combined failure mode of overlying rock collapse and fault slip. By analyzing the goaf failure mode under the influence of the dip angle, the thickness of the ore body, and the location of the ore-controlling fault, it was found that the penetrative failure modes of fault slip (Figure 13a) and roof collapse (Figure 13b) are most likely to occur when mining the top pillar of the submarine metal mine.

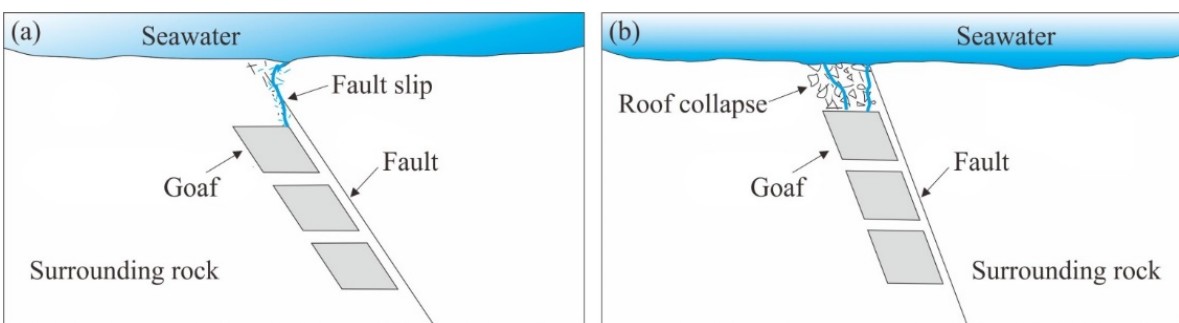

**Figure 13.** Potential penetrative failure modes. (**a**) fault slip; (**b**) roof collapse.

### 6. Conclusions

(1) Subsidence deformation mainly occurred in the goaf roof and hanging wall; the maximum subsidence occurred in the ore body between the goaf and fault, and the subsidence center formed on the hanging wall surface. The footwall rock mass was uplifted near the goaf, and the uplift center formed on the top surface of the ore body. The surrounding rock deformation was significantly different on the two sides of the fault. The horizontal displacement of the surrounding rock mainly occurred in the area that was greatly affected by the redistribution of the stress, and the displacement pointed toward the goaf.

(2) The deep ore body was mainly controlled by the vertical ground stress; its deformation failure mainly extended in the direction of the goaf, and the activity along the fault was relatively weak. The shallow ore body mainly underwent deformation failure along the strike of the fault, which more easily induced fault instability. The physical model tests determined that the critical mining upper limit in the Xinli mine was 50 m, and fault slip occurred when this critical value was exceeded.

(3) As the dip angle and thickness increased, the critical upper mining limit increased, and the failure mode changed from fault slip to roof collapse. When the fault was located in the foot wall, the critical upper mining limit increased as the distance between the fault and the ore body increased, and the failure mode of the goaf was fault slip. When the fault was located in the hanging wall, the final failure mode of the goaf changed to a combined failure mode of overlying rock collapse and fault slip.

There are still several limitations associated with this study. In the laboratory physical model tests, the real water pressure was not applied to the top of the ore body according to the actual situation; moreover, the model construction and monitoring design were also inadequate. Therefore, ideal test data could not be obtained. In addition, only the three main influencing factors, i.e., the dip angle, the thickness of the ore body, and the location of the fault, were considered; the influences of the seawater pressure and the thickness of the Quaternary sediments were ignored. In the experimental design, only the influence of a single controlling factor was considered, and the combined influence of multiple factors on the deformation and failure of the surrounding rock were not considered in accordance with the actual situation. Therefore, further research should be carried out in order to reduce these uncertainties.

**Author Contributions:** Data curation, G.L.; formal analysis, Z.W. and J.G.; methodology, G.L. and Z.W.; software, G.L. and J.L.; writing—original draft, G.L.; writing—review and editing, G.L., J.G. and F.M.; experiments, G.L., J.L., and Y.S. All authors have read and agreed to the published version of the manuscript.

**Funding:** This research was supported by the National Science Foundation of China (Grant Nos. 41831293 and 42072305).

**Acknowledgments:** The authors would like to express their sincere gratitude to Sanshandao Gold Mine for their data support. In addition, the authors are grateful to assigned editors and anonymous reviewers for their enthusiastic help and valuable comments which have greatly improved this paper.

**Conflicts of Interest:** The authors declared that they have no conflicts of interest regarding this study. We declare that we do not have any commercial or associative interests that represent a conflict of interest in connection with the paper submitted.

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
