# Peer review of "A Case Study on Deformation Failure Characteristics of Overlying Strata and Critical Mining Upper Limit in Submarine Mining"

_water, doi:10.3390/w14162465_

Round 1
Reviewer 1 Report
The deformation failure characteristics of the overlying strata and the critical upper mining limit in submarine mining were studuied by physical and numerical test, some meaningful conclusions were obtained. Some suggestions are shown as follows:
(1) "the critical upper mining limit was calculated " was proposed in Abstract, the calculated method was not metioned in the text.
(2) The Figure 5 shows that the broken of the physical model mainly occurred on the surface, whether it can be fully reflected critical upper mining limit.
(3) The biggest difference between underground mining and seabed mining is the influence of seawater, how the factor of seawater considered by authors in the research.
Reviewer 2 Report
Comments:
Aiming at China encountered three mountain island sea gold mining ore rock fracture and damage after being dangerous sea water pouring into the practical engineering problems, Firstly, the simulated ore is configured according to the in-situ ore sampling data of Sanshandao. On the other hand, based on the actual seafloor mining area, the physical model of ore mining is established, and the deformation failure characteristics of overlying strata and critical mining upper limit are obtained. Finally, based on the similarity criterion, the numerical model of mining ore is further established and the multi-parameter sensitivity analysis is carried out. The failure modes under different parameters are obtained by numerical calculation.
The workload of this paper is large, the engineering significance is good, but the innovation point is insufficient, the theoretical research is less, in addition, the logical relationship of this paper is not clear enough. If the author can deal with the following questions, it may be considered for publication.
1. The abstract and conclusion of the article need to be concise, without highlighting key points.
2. The introduction needs to be more logical. This paper mainly introduces the necessity of this research, the background of the project is introduced more, but the basic work of the research is less listed. Authors should revise the abstract, and the work of others should be added.
3. The title of the article is not consistent with the content of the article. The concept of the critical upper mining limit in submarine mining was not mentioned in the numerical simulation part of the article, which was only mentioned in the experimental analysis of the physical model.
4. Although experimental and numerical studies have been carried out in this paper, there is a lack of introduction to the relationship between them, and the correlation between numerical methods and experimental methods is not good. The authors need to provide the main parameters for numerical verification of the experimental results, which should be emphasized in the abstracts and articles.
5. The analysis of the results of physical experiments and numerical calculations is relatively simple, and the mechanical analysis is not strong. The author should increase the theoretical analysis of physical laws.
6. The similarity theory of simplified physical test model needs to be supplemented.
7. The graph in the article is not clear. Please modify each graph.
8. There are some details wrong with the article itself:
1) The data in 3.1 are not consistent with the data in Fig.2;
2) The design value and Similar Material values of Table3 are wrong, please recalculate with reference to the similar relationship
3) The original picture of the test instrument and the field experiment photo should be added to the physical experiment in the article.
